# Adherence and clinical outcomes for twice-daily versus once-daily dosing of non-vitamin K antagonist oral anticoagulants in patients with atrial fibrillation: Is dosing frequency important?

Hui-Jeong Hwang[1], Il Suk Sohn[1]*, Eun-Sun Jin[1], Yoon-Jong Bae[2]

1 Department of Cardiology, Kyung Hee University College of Medicine, Kyung Hee University Hospital at Gangdong, Seoul, Korea, 2 Data Science Team, Hanmi Pharm. Co., Ltd, Seoul, Korea

* issohn@khu.ac.kr

## Abstract

**Data Availability Statement:** All relevant data are within the paper and its Supporting information files. In addition, all raw data have been publicly

### Background

Twice-daily dosing of non-vitamin K antagonist oral anticoagulants (NOACs) may reduce drug adherence compared with once-daily dosing of NOACs in patients with atrial fibrillation (AF), thus worsening clinical outcomes. We evaluated adherence to apixaban and dabigatran requiring twice-daily dosing compared with edoxaban or rivaroxaban with a once-daily dosing regimen and the subsequent clinical outcomes in patients with AF.

### Methods

Adherence to each NOAC and outcomes were compared between patients who were diagnosed with AF and initiated NOACs between 2016 and 2017 using Korean claims data. High adherence was defined as the proportion of days covered (PDC) of the index NOAC ≥80%. The clinical outcomes included stroke, acute myocardial infarction, death, and composite outcome.

### Results

A total of 33,515 patients were analyzed (mean follow-up, 1.7 ± 1.3 years). The proportion of patients with high adherence to NOACs was 95%, which did not significantly differ according to the dosing regimen. The mean PDC for NOACs was as high as ~96%, which was the highest for apixaban users, intermediate for edoxaban or rivaroxaban users, and lowest for dabigatran users, regardless of the dosing regimen. Adverse outcomes in low adherence patients for each NOAC were higher than that of high adherence patients, regardless of the dosing frequency.

available at Korean National Health Insurance Service (NHIS) Bigdata Hub (https://nhiss.nhis.or. kr).

**Funding:** The study was supported by grants from the Korean Cardiology Research Foundation (No.202003-03). The funders had no role in study design, data collection and analysis, decision to publish, or preparation of the manuscript.

**Competing interests:** None declared.

## Conclusions

Adherence between once- and twice-daily dosing NOACs in patients with AF was high and similar among both dosing regimens. Patients with low NOAC adherence had poorer clinical outcomes, regardless of the dosing frequency.

## Introduction

Atrial fibrillation (AF) is one of the most common risk factors for systemic ischemic events, especially increasing the risk of stroke by five-fold [1]; thus, anticoagulation therapy has been recommended for patients with AF who are at a higher risk for stroke. Non-vitamin K antagonist oral anticoagulants (NOACs), rather than warfarin, have been widely used because they do not require therapeutic monitoring and have a lower bleeding risk than warfarin [1–4]. However, the use of NOACs in patients with low adherence may be more harmful because the half-life of NOACs in the plasma is shorter than that of warfarin.

Reducing dosing frequency is known to improve drug adherence. However, adherence to NOACs may be affected by several other factors, including drug-specific side effects, pill size, cost, interactions with concomitant medications, and social and educational status. Furthermore, NOACs have short half-lives and their peak-to-trough ratio is lower in drugs requiring twice-daily dosing. Therefore, the therapeutic coverage of NOACs with a twice-daily dosing regimen may be superior compared with that of once-daily dosing NOACs in patients with low adherence [5]. Consequently, we compared adherence to apixaban and dabigatran that require twice-daily dosing with edoxaban or rivaroxaban (Edox/Riva) that require once-daily dosing, and assessed the subsequent clinical outcomes in patients with AF.

## Methods

### Data sources and study population

This was a retrospective observational study that used the Korean National Health Insurance Service database (NHIS), which provides demographic and medical information of the Korean population covered by medical care, including information on the following: the International Classification of Disease-10th Revision-Clinical Modification (ICD-10-CM) diagnostic codes, prescriptions, procedure codes for inpatient and outpatient visits, and mortality data. We recruited patients who were first diagnosed with non-valvular AF between January 2016 and December 2017 and initiated treatment with NOACs, including apixaban, dabigatran, and Edox/Riva (Fig 1). The date of the first NOAC prescription was defined as the index date. The standard-dose NOACs were defined as 5 mg apixaban and 150 mg dabigatran twice daily and 60 mg edoxaban and 20 mg rivaroxaban once daily. The low-dose NOACs were defined as 2.5 mg apixaban and 110 mg dabigatran twice daily and 30 mg edoxaban and 15 mg rivaroxaban once daily. Patients who were prescribed very low doses of NOACs (e.g., 15 mg edoxaban or 10 mg rivaroxaban) or once-daily dosing of apixaban or dabigatran were excluded. Patients who took warfarin before the index date or NOACs before 2016 were also excluded. Patients aged <20 years and those with a history of end-stage renal disease, dialysis, hyperthyroidism, liver failure including liver cirrhosis, or malignancy were excluded. Patients who had a gap of more than 60 days in the available index drug supply or those who were switched to other anticoagulants were excluded from the analyses. Patients were followed-up until the end of December 2019 for data collection. The hospital Ethics Committee approved this study and

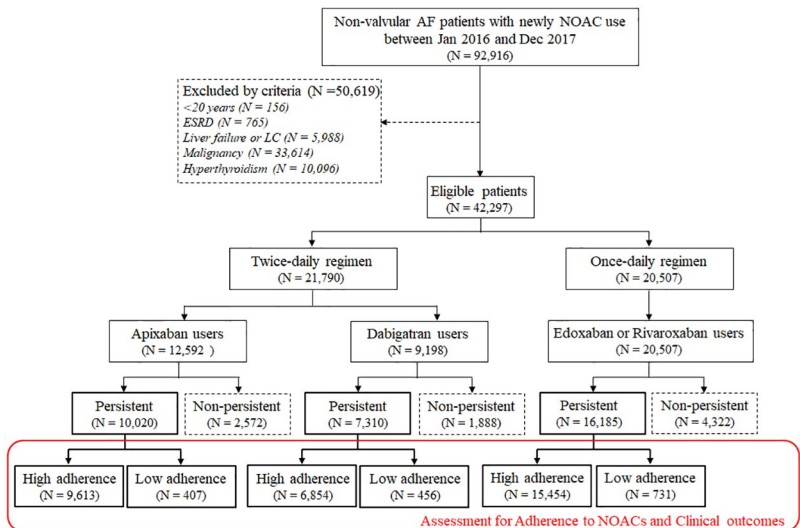

**Fig 1. Study flowchart.** AF, atrial fibrillation; NOAC, non-vitamin K antagonist oral anticoagulant; ESRD, end-stage renal disease; LC, liver cirrhosis.

waived the requirement for informed consent from the patients, because claim data were provided in anonymized and de-identified form (KHNMC 2020-06-016). Data supporting this study are publicly available at the NHIS webpage (http://nhiss.nhis.or.kr).

## Adherence and other variables

Information on NOAC was collected based on prescription claim data. Adherence to NOACs was evaluated using the proportion of days covered (PDC) in patients to continue index NOACs, calculated as the total number of doses supplied divided by the total time the drug was prescribed. In accordance with the commonly used definitions in previous studies [6], patients with PDC ≥80% and PDC <80% were classified into high and low adherence groups, respectively. The time-specific adherence for each NOAC was assessed at 6 months, 1 year, 2 years, and 3 years. Detailed definitions of comorbidities, which were based on the ICD-10-CM diagnosis and/or prescription codes, are described in S1 Table. The CHA2DS2-VASc score was calculated to estimate the stroke risk of patients with AF: 1 point each for congestive heart disease, hypertension, age of 65–74 years, diabetes mellitus, female sex, and vascular diseases, including significant coronary artery disease, myocardial infarction, peripheral artery disease, or aortic plaque, and 2 points each for age ≥75 years and prior stroke, transient ischemic attack, or thromboembolism [1].

## Clinical outcomes

Four clinical outcomes were assessed: stroke, acute myocardial infarction (AMI), all-cause death, and composite outcome of stroke, AMI, and all-cause death. The definitions of each outcome are presented in S1 Table. Patients were censored either at the occurrence of each event or at the end of the study period for each outcome.

## Statistical analysis

Statistical analyses were performed using R software version 4.0.3. Continuous variables are expressed as mean ± standard deviation (or median and interquartile range for variables with

skewed data). Categorical variables are expressed as group percentages. Student's *t*-test and analysis of variance were used to compare continuous variables between groups, and the Chi-square test (or Fisher's exact test for cell counts less than 5) was used to compare categorical variables. Post-hoc tests used the Bonferroni method if data met the assumption of homogeneity of variances, and the Games-Howell method if data did not meet the homogeneity of variances assumption. The adverse outcomes between the groups were compared and presented as unadjusted and adjusted hazard ratios (HRs) and 95% confidence intervals (CIs) using univariate and multivariate Cox proportional hazard models in each propensity score-matched population. Covariates including age, sex, comorbidities (hypertension, diabetes, dyslipidemia, heart failure, chronic renal disease, prior vascular diseases, and stroke), CHA2DS2-VASc score, and concomitant medication use (antiplatelet agents, statins, angiotensin-converting enzyme inhibitors or angiotensin-receptor blockers [ACEIs/ARBs], beta blockers, calcium channel blockers [CCBs], and diuretics) were used for balancing the groups in the propensity score-matched model and for adjusting confounding factors in the multivariate Cox proportional hazards model. In the propensity score matching model, 1:5 propensity score matching was conducted between low and high adherence groups because the proportion of low adherence patients was ~5%, whereas 1:1 matching in the comparison between NOAC groups was performed because the number of patients between groups was similar. Propensity score matching for balancing covariates was performed using the nearest-neighbor method. Each matched model was considered well-balanced when standardized differences of all covariates were ≤0.1. The incidence rates for each outcome are expressed as per 100 person-years of follow-up. The incidence of cumulative events according to NOAC adherence is described using Kaplan-Meier plots and compared using the log-rank test. Competing risk analyses for clinical outcomes were conducted based on the Fine and Gray's model, which showed no significant difference in risk analysis when compared to the Cox proportional hazard models.

The association between NOAC adherence and composite outcome was assessed after propensity score matching in subgroups based on age; sex; comorbidities, including hypertension, diabetes mellitus, dyslipidemia, prior myocardial infarction, stroke, heart failure, and chronic kidney disease; CHA2DS2-VASc score; concomitant antiplatelet agents; and dosing of NOACs. Sensitivity analyses were performed as follows: relative composite outcome risk of low adherence groups for each NOAC (1) in a separate propensity score-matched population with 1-, 2-, and 3-year follow-up periods and (2) when a gap in the available index drug supply for persistent users was permitted to ≤30 or ≤180 days. During the stratified follow-up period, the patients were censored at 1, 2, and 3 years after the index date. Statistical significance was set at p <0.05.

## Results

### Clinical characteristics and adherence for NOACs

A total of 33,515 patients who were first diagnosed with non-valvular AF and initiated NOAC treatment between 2016 and 2017 and continued index drugs were finally enrolled and assessed for adherence to NOACs and clinical outcomes (Fig 1). Their clinical characteristics (mean age, 72 ± 11 years; male to female ratio = 53% to 47%) are described in Table 1. In each propensity-score matched population, all covariates were well-balanced. S2–S4 Tables show clinical data and standardized differences before and after propensity-score matching between the low and high adherence groups. S5 Table shows clinical data and standardized differences after propensity-score matching between the NOAC classes in each high- and low- adherence group.

**Table 1. Clinical characteristics in NOAC users.**

| Characteristics | Total | Apixaban[1] | Dabigatran[2] | Edox/Riva[3] | p value | p value between groups | | |
|---|---|---|---|---|---|---|---|---|
| | (N = 33,515) | (N = 10,020) | (N = 7,310) | (N = 16,185) | | G[1] vs G[2] | G[1] vs G[3] | G[2] vs G[3] |
| Age, years | 72.3 ± 10.9 | 73.5 ± 10.8 | 71.3 ± 10.9 | 72.0 ± 10.8 | <0.001 | <0.001 | <0.001 | <0.001 |
| ≥ 65 years | 26,157 (78) | 8,113 (81) | 5,486 (75) | 12,558 (78) | <0.001 | <0.001 | <0.001 | <0.001 |
| Male, n(%) | 17,620 (53) | 4,853 (48) | 4,096 (56) | 8,671 (54) | <0.001 | <0.001 | <0.001 | <0.001 |
| Medical history, n(%) | | | | | | | | |
| hypertension | 26,883 (80) | 8,069 (81) | 5,800 (79) | 13,014 (80) | 0.102 | 0.054 | 0.822 | 0.059 |
| diabetes | 8,541 (25) | 2,630 (26) | 1,841 (25) | 4,070 (25) | 0.096 | 0.118 | 0.048 | 0.948 |
| dyslipidemia | 19,952 (60) | 6,067 (61) | 4,513 (62) | 9,372 (58) | <0.001 | 0.115 | <0.001 | <0.001 |
| myocardial infarction | 2,941 (9) | 1,049 (10) | 564 (8) | 1,328 (8) | <0.001 | <0.001 | <0.001 | 0.205 |
| stroke | 10,436 (31) | 3,471 (35) | 2,605 (36) | 4,360 (27) | <0.001 | 0.176 | <0.001 | <0.001 |
| thromboembolism | 1,748 (5) | 492 (5) | 382 (5) | 874 (5) | 0.237 | 0.361 | 0.086 | 0.594 |
| heart failure | 15,391 (46) | 4,778 (48) | 3,202 (44) | 7,411 (46) | <0.001 | <0.001 | 0.003 | 0.005 |
| CKD | 1,377 (4) | 563 (6) | 224 (3) | 590 (4) | <0.001 | <0.001 | <0.001 | 0.025 |
| CHA2DS2-VASc | 3.5 ± 1.9 | 3.8 ± 1.8 | 3.6 ± 1.9 | 3.4 ± 1.8 | <0.001 | <0.001 | <0.001 | <0.001 |
| 0–1, n(%) | 4,473 (13) | 1,042 (10) | 982 (13) | 2,449 (15) | <0.001 | <0.001 | <0.001 | <0.001 |
| 2–3, n(%) | 12,979 (39) | 3,644 (36) | 2,710 (37) | 6,625 (41) | <0.001 | 0.350 | <0.001 | <0.001 |
| ≥4, n(%) | 16,063 (48) | 5,334 (53) | 3,618 (49) | 7,111 (44) | <0.001 | <0.001 | <0.001 | <0.001 |
| Medications, n(%) | | | | | | | | |
| low dosing NOAC | 16,296 (49) | 5,089 (51) | 4,523 (62) | 6,684 (41) | <0.001 | <0.001 | <0.001 | <0.001 |
| antiplatelet agent | 12,917 (39) | 3,688 (37) | 2,655 (36) | 6,574 (41) | <0.001 | 0.711 | <0.001 | <0.001 |
| statin | 20,212 (60) | 6,237 (62) | 4,572 (63) | 9,403 (58) | <0.001 | 0.691 | <0.001 | <0.001 |
| ACEI/ARB | 19,152 (57) | 5,732 (57) | 4,115 (56) | 9,305 (57) | 0.233 | 0.232 | 0.653 | 0.088 |
| beta blocker | 16,291 (49) | 5,076 (51) | 3,514 (48) | 7,701 (48) | <0.001 | 0.001 | <0.001 | 0.489 |
| CCB | 14,156 (42) | 4,388 (44) | 3,049 (42) | 6,719 (42) | 0.002 | 0.006 | <0.001 | 0.786 |
| diuretics | 3,417 (10) | 1,156 (12) | 668 (9) | 1,593 (10) | <0.001 | <0.001 | <0.001 | 0.094 |

NOAC, non-vitamin K antagonist oral anticoagulant; Edox/Riva, edoxaban or rivaroxaban; G[1], apixaban; G[2], dabigatran; G[3], Edox/Riva; CKD, chronic kidney disease; ACEI/ARB, angiotensin-converting enzyme inhibitor or angiotensin-receptor blocker; CCB, calcium channel blocker

The proportion of high adherence patients with PDC ≥80% was 95% of NOAC users during overall period (S2 Table). Patients with low adherence to NOACs had a lower incidence of hypertension, diabetes mellitus, and dyslipidemia and a higher incidence of prior thromboembolism, heart failure, and chronic kidney disease. They also had higher CHA2DS2-VASc scores, less statin use, and more diuretic use. The time-specific proportion of high adherence patients did not significantly differ between NOACs requiring once- and twice-daily dosing (Fig 2). The time-specific PDC in NOAC users was 96.1 ± 9.8% at 6-months, 96.3 ± 9.3% at 1-year, 96.3 ± 9.2% at 2-years, and 96.3 ± 9.3% at 3-years (Fig 3, p = 0.01 between the 6-month and 1-year intervals, p was non-significant for all other time intervals), which was not significantly different between patients with once- and twice-daily dosing regimen. The mean PDC for NOACs was the highest for apixaban users, intermediate for Edox/Riva users, and lowest for dabigatran users at all time intervals.

## Adherence and clinical outcomes for each NOAC

The mean follow-up duration was 1.7 ± 1.3 years. The incidence rate of the composite outcome of stroke, AMI, and death was 7.1 (95% CI, 6.9–7.3) per 100 person-years. The cumulative incidence of stroke, AMI, death, and composite outcome was concomitantly higher in patients

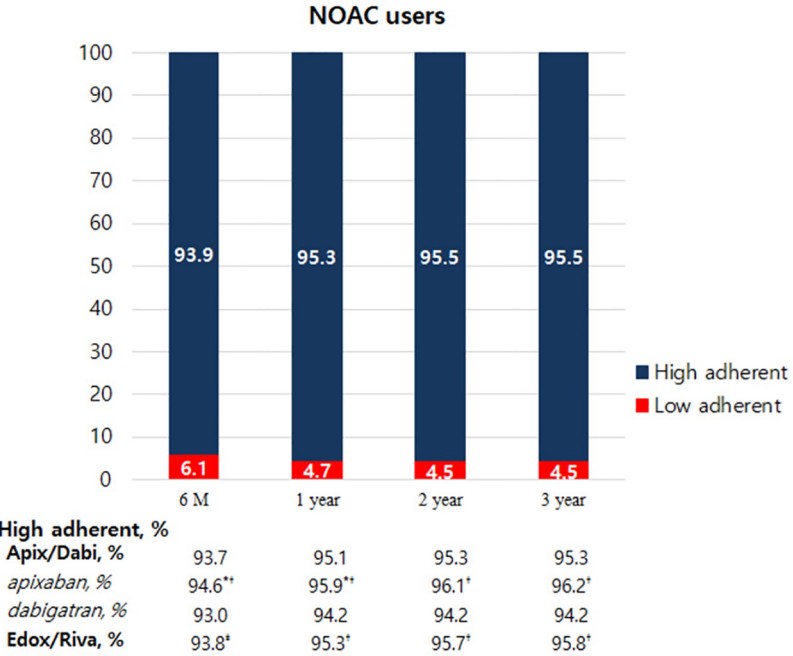

**Fig 2. Time-specific proportions of high and low adherence patients for NOACs.** NOAC, non-vitamin K antagonist oral anticoagulant; M, month; Y, year; Apix/Dabi, apixaban or dabigatran; Edox/Riva, edoxaban or rivaroxaban; *p value <0.05 compared with Edox/Riva users; †p value < 0.001 compared with dabigatran users; ‡p value < 0.05 compared with dabigatran users.

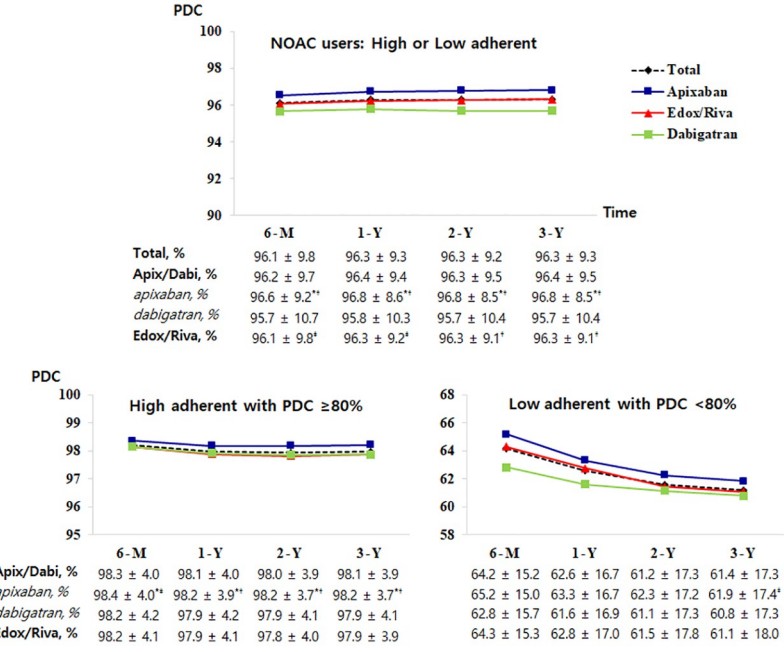

**Fig 3. Time-specific PDC for each NOAC.** PDC, proportion of days covered; NOAC, non-vitamin K antagonist oral anticoagulant; M, month; Y, year; Apix/Dabi, apixaban or dabigatran; Edox/Riva, edoxaban or rivaroxaban; *p value < 0.001 compared with Edox/Riva users; †p value < 0.001 compared with dabigatran users; ‡p value < 0.01 compared with dabigatran users.

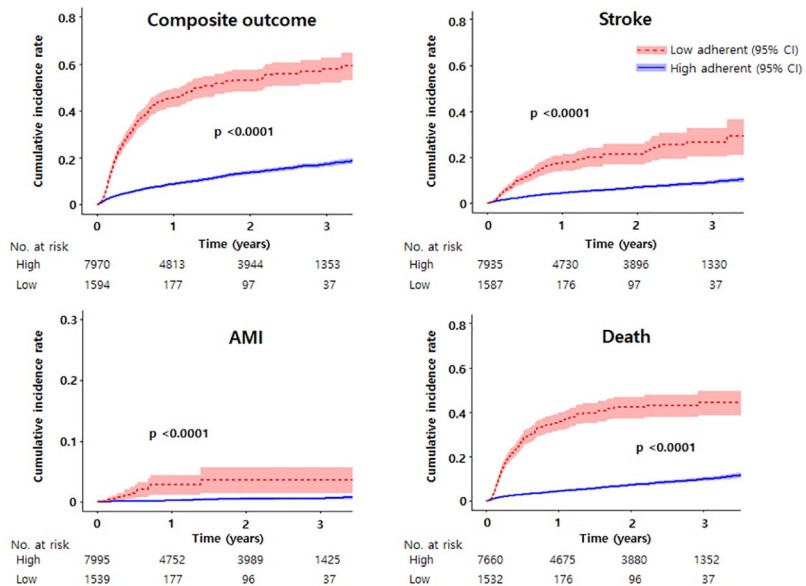

**Fig 4. The cumulative incidence curves of clinical outcomes for patients with high and low adherence to NOACs: Composite outcome, stroke, acute myocardial infarction (AMI), and death.** CI, confidence interval.

with low NOAC adherence than in those with high adherence (Fig 4, p <0.001 for each outcome). Patients with low adherence to each NOAC had a higher HR for each outcome than high adherence patients in each propensity score-matched population (Table 2). However, AMI risk in dabigatran users was not significantly different between patients with high and low adherence (p = 0.292). Each outcome was similar between patients with once- and twice-daily dosing regimens in each group, with either high or low adherence to NOACs (Fig 5). However, the composite outcome risk in patients with low adherence to dabigatran was lower than that of low adherence patients to apixaban (adjusted HR [95% CI], 0.69 [0.51–0.94]; p = 0.018). The stroke risk of patients with high adherence to apixaban was lower than that of patients with high adherence to once-daily dosing NOACs (adjusted HR [95% CI], 0.80 [0.71–0.89]; p <0.001] or dabigatran (adjusted HR [95% CI], 0.85 [0.73–0.99]; p = 0.036).

## Sensitivity and subgroup analysis

The HR for composite outcome in patients with low adherence to each NOAC was concomitantly higher than that of high adherence patients in separate propensity score-matched populations with 1-, 2-, and 3-year follow-up periods (S6 Table, p <0.001 for all outcomes). Similarly, patients with low adherence to each NOAC had a higher composite outcome risk than high adherence patients when a gap in available index drug supply for persistent users was permitted to ≤30 days (adjusted HR [95% CI], 7.40 [6.09–9.00]; p <0.001) and ≤180 days (adjusted HR [95% CI], 3.81 [3.53–4.11]; p <0.001).

The composite outcome risk was significantly higher in patients with low adherence to NOACs compared to that of high adherence patients in the subgroup analysis based on age, sex, medical history of hypertension, diabetes mellitus, dyslipidemia, myocardial infarction, stroke, heart failure, chronic kidney disease, CHA2DS2-VASc score, use of antiplatelet agents, and dosing of NOACs (S1 Fig, p <0.001 for all subgroups).

**Table 2. Event and event rate in NOAC users in the overall study population and the comparative risks for clinical outcomes of patients with low adherence compared with those with high adherence in propensity score-matched population.**

| Clinical outcomes | Overall population | | | Propensity score-matched population[‡] | | | |
|---|---|---|---|---|---|---|---|
| | N* | Events* | Event rate per 100 PY[†] | Unadjusted HR (95% CI) | p value | Adjusted HR[§] (95% CI) | p value |
| **Total** | | | | | | | |
| Composite outcome | 31,921/1,594 | 3,694/424 | 7.1 (6.5/55.9) | 5.52 (4.89–6.23) | <0.001 | 5.62 (4.96–6.37) | <0.001 |
| Stroke | 31,913/1,587 | 1,823/130 | 3.4 (3.2/17.4) | 3.59 (2.93–4.40) | <0.001 | 3.81 (3.10–4.68) | <0.001 |
| AMI | 31,918/1,539 | 152/16 | 0.3 (0.3/2.4) | 7.48 (4.04–13.83) | <0.001 | 9.53 (4.96–18.34) | <0.001 |
| Death | 31,905/1,532 | 1,719/278 | 3.6 (3.2/40.8) | 7.48 (6.42–8.71) | <0.001 | 7.02 (5.97–8.25) | <0.001 |
| **Apixaban** | | | | | | | |
| Composite outcome | 9,613/407 | 1,239/125 | 7.5 (6.9/77.5) | 6.42 (5.11–8.067) | <0.001 | 6.77 (5.34–8.57) | <0.001 |
| Stroke | 9,609/405 | 535/30 | 3.1 (3.0/18.7) | 4.45 (2.88–6.86) | <0.001 | 4.68 (3.01–7.27) | <0.001 |
| AMI | 9,580/397 | 60/4 | 0.4 (0.3/2.5) | 7.29 (2.06–25.82) | <0.001 | 12.16 (3.02–48.90) | <0.001 |
| Death | 9,573/395 | 644/91 | 4.3 (3.8/61.2) | 9.30 (7.06–12.25) | <0.001 | 10.30 (7.66–13.85) | <0.001 |
| **Dabigatran** | | | | | | | |
| Composite outcome | 6,854/456 | 650/85 | 7.3 (6.6/46.4) | 4.62 (3.54–6.04) | <0.001 | 4.35 (3.30–5.72) | <0.001 |
| Stroke | 6,851/455 | 331/32 | 3.6 (3.4/17.5) | 3.61 (2.38–5.46) | <0.001 | 3.61 (2.37–5.48) | <0.001 |
| AMI | 6,866/439 | 31/3 | 0.3 (0.3/1.7) | 3.02 (0.77–11.90) | 0.113 | 2.19 (0.51–9.38) | 0.292 |
| Death | 6,861/438 | 288/50 | 3.5 (3.0/30.9) | 6.39 (4.47–9.14) | <0.001 | 5.04 (3.42–7.41) | <0.001 |
| **Edox/Riva** | | | | | | | |
| Composite outcome | 15,454/731 | 1805/214 | 6.8 (6.2/51.7) | 5.22 (4.41–6.19) | <0.001 | 5.24 (4.39–6.25) | <0.001 |
| Stroke | 15,453/727 | 957/68 | 3.5 (3.3/16.8) | 3.23 (2.45–4.26) | <0.001 | 3.36 (2.53–4.46) | <0.001 |
| AMI | 15,472/703 | 61/9 | 0.3 (0.2/2.7) | 8.79 (3.82–20.23) | <0.001 | 12.42 (4.99–30.93) | <0.001 |
| Death | 15,471/699 | 787/137 | 3.3 (2.8/37.2) | 8.49 (6.79–10.61) | <0.001 | 8.29 (6.51–10.55) | <0.001 |

NOAC, non-vitamin K antagonist oral anticoagulant;

*crude number and events of patients with high adherence/low adherence;

[†]incidence rate of total NOAC users (high adherence users/low adherence users);

[‡]matching covariates, including demographics, CHA2DS2-VASc score, and medical information data between high and low adherence users;

[§]hazard ratio (HR) after adjusting for the aforementioned covariates using Cox proportional hazard models; PY, person-years; CI, confidence interval; AMI, acute myocardial infarction; Edox/Riva, edoxaban or rivaroxaban

## Discussion

In this study, we found that adherence to each NOAC was not significantly dependent on the dosing regimen, and the clinical outcomes for patients with low adherence were poor regardless of the dosing frequency.

Adherence to NOACs varies by race and nation [3, 7–9]. In a meta-analysis of 48 observational studies with real-world data on NOAC adherence in patients with AF, the 1-year mean PDC or medication possession ratio (MPR) and high adherence proportion assessed using PDC/MPR ≥80% for NOACs were approximately 80% (95% CI, 72–86%) and 68% (95% CI, 62–74%), respectively, which were significantly higher in European cohorts than in North American cohorts [7]. Additionally, a recent observational study conducted in Japan [9] showed higher NOAC adherence (1-year mean PDC, 95%; high adherence proportion, 95%) than the aforementioned Western country-based data [7], which is consistent with our study based on Korean claims data.

Several previous studies have shown that NOAC adherence requiring twice-daily dosing was lower than that of once-daily dosing [8, 10]. However, some studies individualizing each NOAC showed that the adherence to apixaban was similar to that of rivaroxaban, whereas

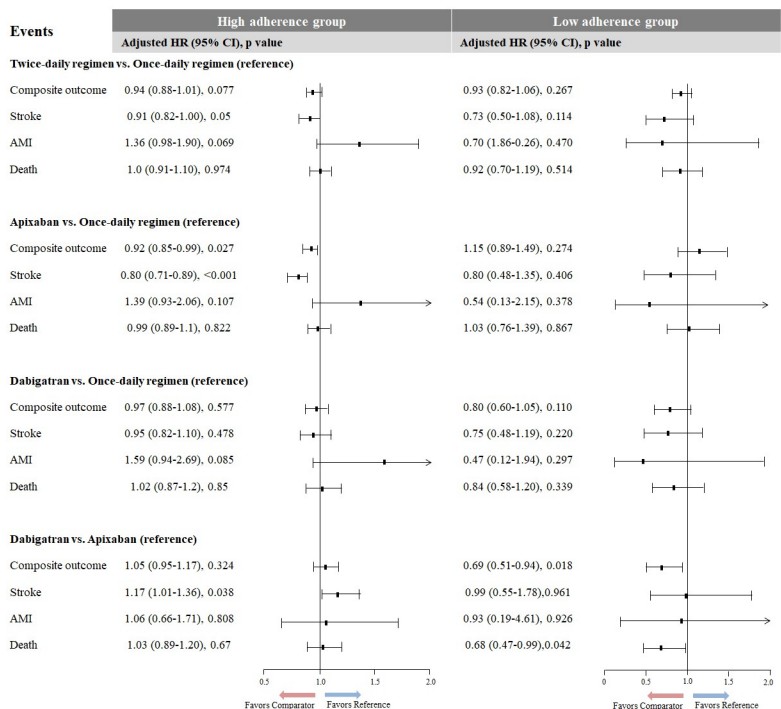

**Fig 5. Comparative clinical outcomes of each NOAC in high and low adherence groups.** NOAC, non-vitamin K antagonist oral anticoagulant; HR, hazard ratio; CI, confidence interval; AMI, acute myocardial infarction.

dabigatran had lower adherence than apixaban or rivaroxaban [7, 11], although both apixaban and dabigatran require twice-daily dosing. In the present study, the mean PDC for apixaban was higher than that of Edox/Riva, and the mean PDC for dabigatran was the lowest of all NOACs. This may be because NOAC adherence is more affected by drug-specific properties than their dosing regimens, including common side effects such as dyspepsia [9, 12, 13] or interaction with verapamil in dabigatran users [1].

Because the NOAC effect is rapidly eliminated and twice-daily dosing maintains continuous plasma drug levels of less hazardously high peaks and low troughs when compare to once-daily dosing, twice-daily dosing of NOACs has been believed to theoretically better prevent ischemic events when several dosing is omitted [14]. However, a real-world study by Alberts et al. demonstrated that the increase in stroke risk in patients with low NOAC adherence was not associated with the dosing frequency of NOACs [10]. Likewise, in the present study, clinical outcomes, including stroke, were not significantly different between the once- and twice-daily dosing regimens in patients with low adherence. Separately from this, the efficacy of each NOAC was somewhat different depending on drug adherence: the composite outcome in patients with low adherence to dabigatran was better than that of patients with low adherence to apixaban, whereas high adherence to apixaban was associated with better stroke protection when compared to high adherence to Edox/Riva or dabigatran. When NOACs were taken optimally, the superior efficacy of apixaban in stroke prevention compared with rivaroxaban and dabigatran has been reported [15], which is also consistent with our study. However, the difference in the efficacy of each NOAC in low adherence patients has not been clarified. Although the definite mechanisms cannot be explained, better efficacy in patients with low adherence to dabigatran compared to apixaban may be due to drug-specific properties. Indeed,

dabigatran showed similar stroke and death rates compared with warfarin even when a low-dose was administered [16], whereas apixaban and rivaroxaban had a higher death rate in low-dose in Danish [17] and UK [18] studies. However, this finding does not guarantee the safety of low adherence to dabigatran because the clinical outcomes of patients with low adherence to dabigatran are much worse than patients with high adherence.

In this study, patients with low adherence to apixaban and Edox/Riva had significantly more AMI events than those with high adherence. This finding may be a natural result because anticoagulation therapy is useful in preventing AMI, rather than the use of anti-platelet agents [19–21]. In addition, recent studies on adherence to NOACs and AMI risk [22, 23] showed similar results. On the other hand, the efficacy of AMI protection in dabigatran users was not statistically significant between high and low adherence patients in our study. However, this does not mean that dabigatran is not useful for protection against AMI events. Indeed, the AMI incidence rate in high adherence patients was similar in all NOCA classes.

In the present study, patients with low adherence to NOACs had a higher risk of death than those with high adherence. Moreover, their death rate must be very high, even though mortality from cardiovascular or embolic events due to discontinuation of NOAC was considered. This may be because several negative factors that are not generally considered, such as physical fragility [24], economic poverty, and other social factors [25], lead to low adherence to NOACs and subsequent adverse outcomes. Therefore, adherence to NOACs should be considered as a predictive parameter of the overall adverse situation of patients, but not simply as a marker to assess the laziness or neglect of patients.

This study has some limitations. First, the index drug adherence was based on prescription claims data; thus, it does not guarantee that the patients actually took the index drug. However, it allows for a good estimate of drug adherence through prescription patterns. Second, the permissible gap in the available index drug supply for defining persistence to NOACs varied widely from 14 to 365 days in previous studies, in which the most commonly used definition was a 60-day gap [7]. Assessment of clinical outcomes according to strict definition results in many excluded patients, which may have low utility in clinical settings. Therefore, we allowed the persistence of a 60-day gap to assess adherence to the index drugs and conducted sensitivity tests using different permissible gaps of $\leq 30$ and $\leq 180$ days. Consequently, the outcome risk of patients with low adherence to NOACs was consistently higher for all permissible gaps. Third, major bleeding events were not considered as clinical outcomes. In the claim data, it is difficult to distinguish whether a bleeding event is a consequence of NOAC adherence or a cause of lower NOAC adherence. Fourth, we classified edoxaban and rivaroxaban as once-daily dosing regimen NOACs but did not individualize each. This is because the drug-specific properties of edoxaban and rivaroxaban have not been defined. Furthermore, this study focused on evaluating the differences in drug adherence according to dosing regimens. Fifth, we did not evaluate whether patients administered low-dose NOACs received on- or off-label dosing.

## Conclusions

Adherence for each NOAC in patients with AF was similar among both dosing regimens and high, which may be due to drug-specific properties rather than the dosing regimens. In addition, patients with low adherence to NOACs had poorer clinical outcomes, regardless of the dosing frequency. Thus, individualized education and careful monitoring of medication-taking behaviors and side effects are required to improve NOAC adherence and obtain better outcomes in patients following NOAC treatment.

## Supporting information

**S1 Table. Definitions of atrial fibrillation, comorbidities, medications, and clinical outcomes.**
(DOCX)

**S2 Table. Clinical characteristics according to NOAC adherence before and after propensity score matching for composite outcome.**
(DOCX)

**S3 Table. Baseline characteristics and standardized differences after propensity score matching for composite outcome between patients with high (PDC ≥80%) and low (PDC <80%) adherence to each NOAC.**
(DOCX)

**S4 Table. Baseline characteristics and standardized differences after propensity score matching for stroke, AMI, and death between patients with high (PDC ≥80%) and low (PDC <80%) adherence to each NOAC.**
(DOCX)

**S5 Table. Baseline characteristics and standardized differences after propensity score matching for composite outcome, stroke, AMI, and death between NOAC classes.**
(DOCX)

**S6 Table. Sensitivity analyses: Events and event rates in NOAC users in the overall study population and the comparative composite outcome risks of patients with low adherence compared with those with high adherence in propensity score-matched population.**
(DOCX)

**S1 Fig. Comparative risks for composite outcome of patients with low adherence compared with those with high adherence in subgroups based on age, sex, hypertension, diabetes, dyslipidemia, myocardial infarction, stroke, heart failure, chronic kidney disease, CHA2DS2-VASc score, use of antiplatelet agents, and NOAC dosing.**
(TIF)

## Author Contributions

**Conceptualization:** Hui-Jeong Hwang.

**Data curation:** Hui-Jeong Hwang, Il Suk Sohn, Eun-Sun Jin, Yoon-Jong Bae.

**Formal analysis:** Hui-Jeong Hwang, Il Suk Sohn, Yoon-Jong Bae.

**Funding acquisition:** Hui-Jeong Hwang.

**Investigation:** Hui-Jeong Hwang.

**Methodology:** Hui-Jeong Hwang, Il Suk Sohn, Eun-Sun Jin, Yoon-Jong Bae.

**Project administration:** Hui-Jeong Hwang.

**Resources:** Hui-Jeong Hwang, Yoon-Jong Bae.

**Software:** Hui-Jeong Hwang, Yoon-Jong Bae.

**Supervision:** Hui-Jeong Hwang, Il Suk Sohn, Eun-Sun Jin.

**Validation:** Hui-Jeong Hwang, Yoon-Jong Bae.

**Visualization:** Hui-Jeong Hwang, Il Suk Sohn, Yoon-Jong Bae.

**Writing – original draft:** Hui-Jeong Hwang.

**Writing – review & editing:** Il Suk Sohn, Eun-Sun Jin.

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
