## [Decision Letter · Decision Letter 0]

24 Nov 2022

PONE-D-22-19616Adherence and Clinical Outcomes for Twice-Daily Versus Once-Daily Dosing of Direct Oral Anticoagulants in Patients with Atrial Fibrillation: Is Dosing Frequency Important?PLOS ONE

Dear Dr. Sohn,

Thank you for submitting your manuscript to PLOS ONE. After careful consideration, we feel that it has merit but does not fully meet PLOS ONE’s publication criteria as it currently stands. Therefore, we invite you to submit a revised version of the manuscript that addresses the points raised during the review process.

We look forward to receiving your revised manuscript.

Kind regards,

Carmine Pizzi

Academic Editor

PLOS ONE

Journal Requirements:

Reviewers' comments:

Reviewer's Responses to Questions

**Comments to the Author**

1. Is the manuscript technically sound, and do the data support the conclusions?

Reviewer #1: Yes

Reviewer #2: Partly

2. Has the statistical analysis been performed appropriately and rigorously? 

Reviewer #1: Yes

Reviewer #2: Yes

3. Have the authors made all data underlying the findings in their manuscript fully available?

Reviewer #1: Yes

Reviewer #2: Yes

4. Is the manuscript presented in an intelligible fashion and written in standard English?

Reviewer #1: Yes

Reviewer #2: Yes

5. Review Comments to the Author

Reviewer #1: I read the manuscript with interest, also for the practical impact in our daily activity. Adherence to anticoagulation therapy (i.e. DOACs) is critical in patients with non-valvular AF and, the authors have rightly pointed out the need to promote a higher adherence among patients.

Overall, the data of this study constitute an important information about how to properly manage anti thrombotic medication in patients with non-valvular AF

Reviewer #2: The paper of L SUK Sohn discuss the relevant issue of the impact of adherence to DOAC treatment on different drug regimens. They found no impact of once a day vs. twice a day administration in terms of adherence, while in both cases the clinical outcomes were worst in low adherence cohort.

The topic is relevant ad results interesting. However, there are some methodological concerns.

1) We have no data on dose prescription according to smpc. I particular the number of subjects treated with low doses seems quite high in general and this is particularly relevant for apixaban considering the small amount of patients receiving low doses in the registration trial.

2) Authors show only CKD prevalence which is really low for this type of population whereas it has a big impact on clinical outcomes and adherence due to both ischemic and hemorragic events

3) It is not clear the use of propensity score matching because it seems to be used to compare patients with low vs. high adherence especially in table 2 but looking at the methods section and considering the comparrison between different agents in the subgroup of patients with high adherence it is not clear if this comparrison was made on propoensity matched cohorts. If this was done, clinical characteristics of the matched cohorts should be reported otherwise the comparrison between cohorts non directly matched should be omitted.

6. PLOS authors have the option to publish the peer review history of their article (what does this mean?). If published, this will include your full peer review and any attached files.

Reviewer #1: No

Reviewer #2: No

---

## [Author Response · Author response to Decision Letter 0]

6 Dec 2022

Journal Requirements:

Answer) The manuscript complied with the style requirements of PLOS ONE. 

Answer) Because Korean claim data are anonymized and de-identified, patients’ informed consent was waved. We add the fact in the manuscript.

(in manuscript) The hospital Ethics Committee approved this study and waived the requirement for informed consent from the patients, because claim data were provided in anonymized and de-identified form (KHNMC 2020-06-016).

We will update your Data Availability statement to reflect the information you provide in your cover letter. If there are restrictions on publicly sharing data—e.g. participant privacy or use of data from a third party—those must be specified.

Answer) Data cannot be shared publicly because of the provisions of the National Health Insurance Service (NHIS). Korean legal restrictions prohibit authors from making the data publicly available, and the authority implemented the restrictions is NHIS (National Health Insurance Service), one of the government agency of Republic of Korea. NHIS provides limited portion of anonymized data to the researchers for the purpose of the public interest. However, they exclusively provide data to whom made direct contact of the NHIS and agreed to policies of NHIS. Redistribution of the data is not permitted for the researchers. We add the fact in the manuscript.

(in manuscript) Data supporting this study are publicly available at the NHIS webpage (http://nhiss.nhis.or.kr). 

Reviewers' comments:

Reviewer #1: I read the manuscript with interest, also for the practical impact in our daily activity. Adherence to anticoagulation therapy (i.e. DOACs) is critical in patients with non-valvular AF and, the authors have rightly pointed out the need to promote a higher adherence among patients. Overall, the data of this study constitute an important information about how to properly manage anti thrombotic medication in patients with non-valvular AF

Answer) Thank you for your good estimation.

Reviewer #2: The paper of L SUK Sohn discusses the relevant issue of the impact of adherence to DOAC treatment on different drug regimens. They found no impact of once a day vs. twice a day administration in terms of adherence, while in both cases the clinical outcomes were worst in low adherence cohort.

The topic is relevant ad results interesting. However, there are some methodological concerns.

1. We have no data on dose prescription according to smpc. I particular the number of subjects treated with low doses seems quite high in general and this is particularly relevant for apixaban considering the small amount of patients receiving low doses in the registration trial.

Answer) Based on your comments, we added the definition on usage and dosage of standard and low dose NOACs. The percentage of low-dose NOACs was high, ranging from 40 to 60%, which was higher in low-dose dabigatran users than apixaban users (Table 1). Clinical outcomes in the low adherence group were consistently worse in low- and standard- dose NOAC users (Supple Fig 1). 

(in manuscript) The standard-dose NOACs were defined as 5 mg apixaban and 150 mg dabigatran twice daily and 60 mg edoxaban and 20 mg rivaroxaban once daily. The low-dose NOACs were defined as 2.5 mg apixaban and 110 mg dabigatran twice daily and 30 mg edoxaban and 15 mg rivaroxaban once daily. Patients who were prescribed very low doses of NOACs (e.g., 15 mg edoxaban or 10 mg rivaroxaban) or once-daily dosing of apixaban or dabigatran were excluded.

2. Authors show only CKD prevalence which is really low for this type of population whereas it has a big impact on clinical outcomes and adherence due to both ischemic and hemorragic events.

Answer) Patients with end-stage renal disease were excluded in this study (Fig 1). Furthermore, NOAC use is restricted in CKD patients with creatinine clearance < 15-30 mg/min. Thus, CKD rate was low in the study enrollment population (3-6% in Table 1). It is clear that CKD is an important risk factor for clinical outcomes. Thus, CKD was used as a covariate on propensity score and multivariate Cox proportional hazard models for clinical outcome assessment (S2 – 5 Tables).

3. It is not clear the use of propensity score matching because it seems to be used to compare patients with low vs. high adherence especially in table 2 but looking at the methods section and considering the comparison between different agents in the subgroup of patients with high adherence it is not clear if this comparison was made on propensity matched cohorts. If this was done, clinical characteristics of the matched cohorts should be reported otherwise the comparison between cohorts non directly matched should be omitted.

Answer) S5 table shows the standardized differences before and after propensity score matching of clinical characteristics between each NOAC in figure 5 (comparative clinical outcomes of each NOAC in the high and low adherence groups), in which matched clinical data including demographics are skipped due to their high volume. However, as per your recommendation, we revised them in the S5 table. Similarly, S2 – S4 tables show standardized differences and/or clinical data before and after propensity score matching to compare clinical outcomes between the low and high adherence groups (this is, clinical data of propensity score matched population supporting Table 2 results). In S4 table, we added clinical data after propensity score matching between the two groups.

---

## [Editor Report · Decision Letter 1]

9 Mar 2023

Adherence and Clinical Outcomes for Twice-Daily Versus Once-Daily Dosing of Non-Vitamin K Antagonist Oral Anticoagulants in Patients with Atrial Fibrillation: Is Dosing Frequency Important?

PONE-D-22-19616R1

Dear Dr. Sohn,

We’re pleased to inform you that your manuscript has been judged scientifically suitable for publication and will be formally accepted for publication once it meets all outstanding technical requirements.

Kind regards,

Giuseppe Andò, M.D., Ph.D.

Academic Editor

PLOS ONE
---

## [Editor Report · Acceptance letter]

21 Mar 2023

PONE-D-22-19616R1 

Adherence and Clinical Outcomes for Twice-Daily Versus Once-Daily Dosing of Non-Vitamin K Antagonist Oral Anticoagulants in Patients with Atrial Fibrillation: Is Dosing Frequency Important? 

Dear Dr. Sohn:

I'm pleased to inform you that your manuscript has been deemed suitable for publication in PLOS ONE. Congratulations! Your manuscript is now with our production department. 

Kind regards, 

on behalf of

Prof. Giuseppe Andò 

Academic Editor

PLOS ONE